# EIM: An effective solution for improving multi-modal large language models

**Yuting Bai**[1], **Tonghua Su**[1*], **Zixing Bai**[2*]

**1** School of Software, Harbin Institute of Technology, Harbin, China, **2** School of Computer Science and Technology, Fudan University, Shanghai, China

☯ These authors contributed equally to this work.
\* thsu@hit.edu.cn (TS); 2322459556@qq.com (ZB)

**Data availability statement:** All relevant data are within the manuscript.

**Funding:** The author(s) received no specific funding for this work.

## Abstract

Enabling large language models (LLMs) to have multi-modal capabilities, such as vision-language learning, has become a current research hotspot and the next milestone in LLM development with the advent of models like GPT4. The basic structure of current multi-modal LLMs usually includes three parts: the image encoder for extracting visual features, the **s**emantic space **t**ransformation network **ST** for aligning the multi-modal semantic spaces, and LLM for generating text. Current works on multi-modal LLMs primarily focus on enhancing performance by utilizing larger image encoders and LLMs, and designing more complex fine-tuning methods and STs, which results in an escalation of model parameters. In this paper, we propose **EIM**, a novel **e**ffective solution for **i**mproving the performance of **m**ulti-modal large language models from the perspective of training process which reduces the need to introduce new parameters and modify the model structure, and is ignored and less explored in current research. EIM includes corresponding improvement measures in the image encoder, ST, and LLM. To validate EIM, we first apply it to ClipCap and conduct experiments on the COCO Caption dataset. Secondly, we extend EIM to the multi-modal LLMs, such as LLaMA-Adapter and LaVIN, and evaluate them on the ScienceQA dataset. Finally, we also conduct multi-modal chatbot experiments with the EIM enhanced LaVIN and evaluate it on the MME benchmark. The COCO Caption dataset experimental results of $ClipCap_{eim}$, which is a model that applies EIM on the $ClipCap_{small}$, show the 1.75% performance improvement when compared to those of $ClipCap_{large}$, which has 3.13 times the number of parameters of $ClipCap_{eim}$. The experimental results on the ScienceQA dataset and MME benchmark show that EIM can achieve competitive performance with 7B model parameters when compared to the 13B multi-modal LLMs, which confirms the effective performance improvement of EIM for multi-modal LLMs.

## Introduction

In recent years, with the increase in model parameter size, large language models (LLMs), such as the GPT series [1–4] and LLaMA series [5–8], continuously push the upper limit of natural language understanding. The next milestone of LLMs is usually considered to enable

**Competing interests:** The authors have declared that no competing interests exist.

multi-modal capabilities. For instance, GPT4 [4] not only has excellent text comprehension ability but also supports multi-modal data such as images. The trend of enabling LLMs with multi-modal capabilities, such as image captioning and visual Q&A, has stimulated research and development in the field of multi-modal LLMs [9–28].

CLIP [29] has become the de facto fundamental model in the current multi-modal field due to its excellent performance [30–40]. For instance, the combination of CLIP and diffusion models [41–44] can generate stunning images solely based on text instructions [45–47]. With the rapid development of LLMs [9,48], how to combine CLIP and LLMs to support multi-modal tasks, such as visual language generation, has become a research hotspot [9–15,17–20,22–24,26,57,58].

ClipCap [49] is the earliest case to consider combining CLIP and language models, which converts images into text through the three-segment structure: the image encoder CLIP for extracting visual features, the **s**emantic space **t**ransformation network **ST** for aligning the multi-modal semantic spaces, and the language model GPT-2 [2] for generating text. With the development of LLMs [4,6,48,50] and CLIP-like models [29,51–56], the three-segment structural design similar to ClipCap that connects two different modal models through ST has shown strong versatility and has been widely used in recent multi-modal LLMs [11,12,14,19, 22,24]. For instance, LLaMA-Adapter [19] uses LLaMA [5] as the LLM, and PandaGPT [16] uses ImageBind [56] as the image encoder and Vicuna [7,8] as the LLM. Following the three-segment structure, some recent works, like Emu3 [59], Show-o [60], InternVL3 [61], Janus-Pro [62], extend the multi-modal LLM ability from visual understanding to visual understanding and generation by introducing the additional image decoder. However, the training costs of this new structure are too expensive. Its biggest highlight, image generation, is a bit lackluster because it can only generate an image that is the same as the model's input image, lacking practical application scenarios. Therefore, we only consider the three-segment structure in this paper. In summary, current research on multi-modal LLMs is still in its early stage and mainly focuses on using larger image encoders or LLMs to improve the performance.

We summarize the problems and shortcomings of current research in the field of multi-modal LLMs as below:

- Current research on multi-modal LLMs primarily concentrates on enhancing performance by utilizing larger image encoders [29,56] and LLMs [5,7], designing more complex fine-tuning methods like MMA [22] and STs like Q-Former [23], as well as a huge number of high-quality image-text pairs for pre-training ST to align the semantic spaces between vision and language [10–13,25,28], which results in an escalation of model parameters and a surge in training costs.
- Although text features have been proven to help improve the performance of CLIP [31,63–65], there is currently a lack of research cases in the field of multi-modal LLMs that utilize text features of CLIP. Some studies have even suggested that fine-tuning CLIP may be detrimental to downstream tasks training [36,66–68].
- Current research on multi-modal LLMs usually only uses the auto-regressive training objective [11,19,22]. Optimizing multi-modal LLMs from the perspective of the training process is ignored and less explored.

In this paper, we think the training process of multi-modal LLMs should be different from that of traditional autoregressive models due to the multi-modal input rather than uni-modal input. Besides, optimizing the training process can improve the model performance, reduce the need to introduce new parameters and modify the original model structure, be orthogonal to the mainstream solutions, and enable the utilization of text features of CLIP. Therefore,

we propose an **e**ffective solution for **i**mproving **m**ulti-modal LLMs called **EIM** that enhances model performance from the perspective of the training process. Our approach involves utilizing the text features of CLIP and adding contrastive losses on CLIP, ST, and LLM. We conduct ablation studies on ClipCap and quantitatively analyze the effects in detail. Furthermore, we extend EIM to LLaMA-Adapter [19] and LaVIN [22] for validation. The experimental results confirm the effective performance improvement of EIM for multi-modal LLMs. Overall, our paper makes the following contributions:

- We propose three contrastive losses to improve the performance from the perspective of the training process. The proposed losses are used for CLIP, ST, and LLM, respectively. And we further provide a solution that combines these losses to achieve stable performance improvement.
- We propose to enhance CLIP's capability in multi-modal LLMs by introducing textual information. Although CLIP has the ability to extract both image and text features, as it undergoes image-text alignment during the pre-training stage, previous studies primarily focus on utilizing CLIP's visual feature extraction capability while neglecting its text feature extraction capability. To address this limitation, the CLIP text encoder is introduced to encode the textual information and help to guide the CLIP fine-tuning during the training.
- Based on the above improvements, we propose **EIM**, a novel **e**ffective solution for **i**mproving the performance of **m**ulti-modal large language models from the perspective of the training process. We apply EIM on the $ClipCap_{small}$ model and achieve 1.75% performance improvement on the COCO Caption dataset when comparing to the experimental results of $ClipCap_{large}$, which has 3.13 times the number of parameters of $ClipCap_{small}$. Furthermore, we extend EIM to the representative multi-modal LLMs, such as LLaMA-Adapter and LaVIN, and evaluate on the ScienceQA dataset, achieving accuracy improvements of **2.76**% and **2.05**%, respectively. Finally, we apply EIM on LaVIN-7B-lite and evaluate on the MME benchmark, achieving comparable performance when compared to LaVIN-13B.

## Related work

Due to the widely used three-segment structural design in current multi-modal LLMs, we will introduce the related multi-modal LLMs from the image encoder, ST, and LLM.

**Image encoder.** PandaGPT [16] extracts visual features using ImageBind [56]. Compared to CLIP [29], ImageBind supports a wider variety of modal data. Similar to PandaGPT, ImageBind-LLM [28] also uses ImageBind to encode more modal data. The EVA-CLIP series [51–53], which achieve better performance through improving the training techniques of CLIP, are broadly introduced in models like BLIP-2 [23], InstructBLIP [24], MiniGPT-4 [12], Lynx [25], Ziya-Visual [17], and LLaVA-1.5 [10]. Unlike existing works, we improve the performance of CLIP by using contrastive learning, introducing textual information, removing CLIP's output layer, and using the full visual features rather than replacing CLIP with larger image encoders like ImageBind and the EVA-CLIP series.

**ST.** Most of the current multi-modal LLMs, such as LLaVA series [10,11], LLaMA-Adapter [19,20], Otter [15], MultiModal-GPT [18], MiniGPT-4 [13], VisualGLM-6B [69,70], and InstructBLIP [24], use a large number of high-quality image-text pairs for pre-training ST to align the semantic spaces between vision and language, which results in a surge in training costs. Although LaVIN [22] reduces the training costs by Mixture-of-Modality Training without the pre-training stage, there is no alignment design for ST. Therefore, a novel contrastive

loss for ST is proposed in this paper to reduce the training costs, and also take into account the alignment of ST and LLM.

**LLM.** LLaVA [10,11], LLaMA-Adapter [19,20], and LaVIN [21,22] use the LLaMA series [5–8] and use PETL (Parameter-Efficient Transfer Learning) [19–21,21,22,71–79] to optimize the model. Unlike existing works, we introduce contrastive learning to help improve the performance of LLMs rather than solely relying on using PETL and replacing LLMs with larger ones.

## Method

As shown in Fig 1, EIM includes the following improvement measures. Firstly, the CLIP text encoder *CLIP.text* is introduced to encode the textual information and help to guide the CLIP fine-tuning through $loss_{IE}$ during the training. Secondly, the contrastive loss $loss_{ST}$ is proposed to align the visual and text semantics. Finally, the contrastive loss $loss_{LM}$ is proposed to help to guide the text generation.

Following the previous contrastive learning works [49,80–84], there is a queue $K = (k^1, \cdots, k^i, \cdots, k^N)$ to store key vectors for all samples, the positive sample refer to the input sample itself, and the negative samples refer to all other samples in the dataset. The details are

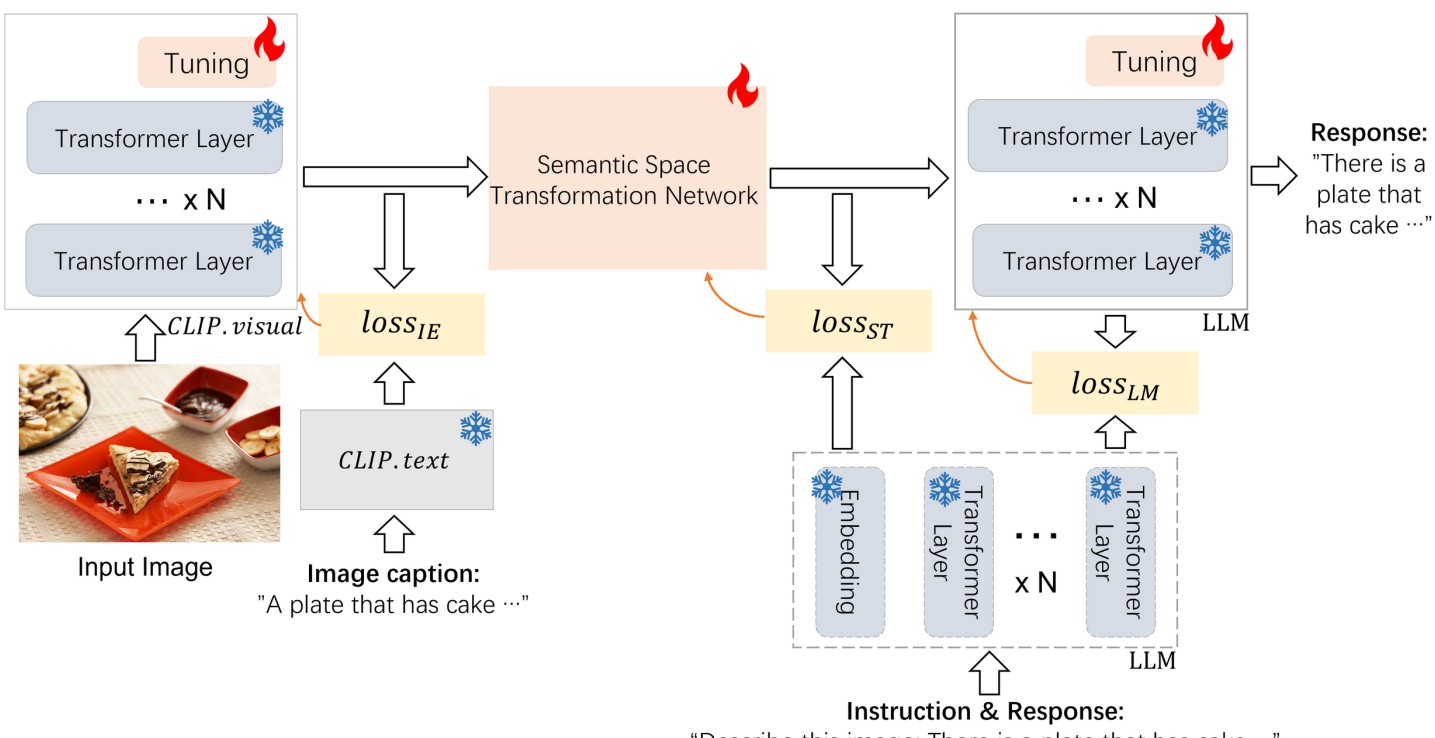

**Fig 1. The overview of EIM. EIM** is an **e**ffective solution for **i**mproving the performance of **m**ulti-modal large language models from the perspective of the training process. EIM includes using three contrastive losses: $loss_{IE}$, $loss_{ST}$ and $loss_{LM}$, and introducing *CLIP.text*. **loss$_{IE}$** is the contrastive **loss** for the **I**mage **E**ncoder *CLIP.visual*. **loss$_{ST}$** is the contrastive **loss** for the **S**emantic **S**pace **T**ransformation Network. **loss$_{LM}$** is the contrastive **loss** for the **LM**. $loss_{IE}$ and $loss_{LM}$ should be used together with fine-tuning methods. There is no limitation on the CLIP and LM fine-tuning methods, which depend on the original model implementations. If the original multi-modal LLMs do not provide the fine-tuning methods for CLIP, we can skip using it or use the prompt-tuning method by default.

shown in Fig 2. The contrastive loss aims to maximize the similarity between the query vector $q^i$ and the key vector $k^i$, while minimizing the similarity between $q^i$ and all other $k^j$, where $j \neq i$.

It is worth noting that in current multi-modal LLMs, the utilization of the output layer of CLIP is detrimental to downstream tasks training due to the differences between the pre-training task and downstream tasks, and the utilization of the partial visual features leads to the issue of losing information. Therefore, the CLIP image encoding capability is improved in this paper by modifying the usage of CLIP features, which includes removing CLIP's output layer and using the full visual features. The CLIP's output layer is a linear layer used to change the hidden dimension to the output dimension, ensuring that the output dimension of the CLIP.visual is consistent with that of the CLIP.text. The full visual features are the [CLS] token and the patch tokens extracted by CLIP.

### Base structure

Fig 3 illustrates the basic structure of multi-modal LLMs, which includes three parts: the image encoder *CLIP.visual* for extracting visual features, ST for aligning the multi-modal semantic spaces, and LLM for generating text.

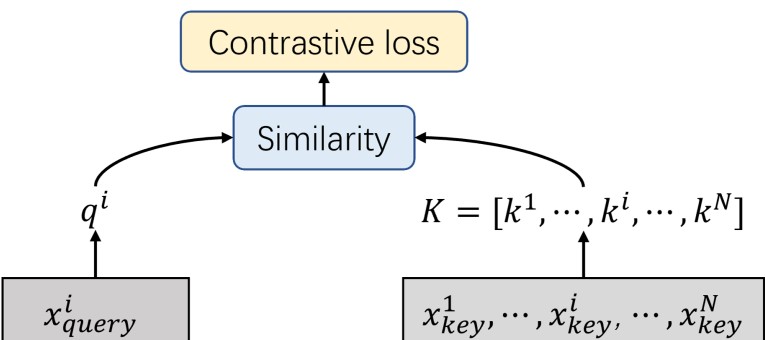

**Fig 2. The overview of contrastive learning.** For $x_{query}^i$, the positive sample is the $x_{key}^i$, the negative samples are $x_{key}^j$, where $j \neq i$. The larger the queue $K$, the richer the visual information and features it can represent. Then, when using queries for comparative learning, the more features of images can be learned. Therefore, previous works usually use all samples to build the queue $K$.

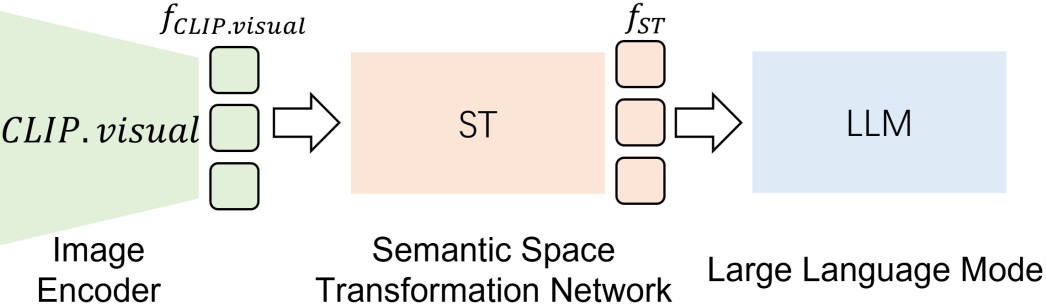

**Fig 3. The overview of multi-modal LLMs.** The structure of multi-modal LLMs contains three modules: the image encoder *CLIP.visual* for extracting visual features, ST for aligning the multi-modal semantic spaces, and LLM for generating text.

Current models usually use the auto-regressive training objective, which is called $loss_{base}$ in this paper. Given the response $(t_1, t_2, t_3, \cdots, t_j, \cdots, t_m)$, and the ST output features $f_{ST}$, the training objective $loss_{base}$ is defined by:

$$loss_{base} = -\sum_{j=1}^{m} logp(t_j | f_{ST}, t_1, \cdots, t_{j-1}).$$

(1)

Here, $m$ represents the length of the response.

## CLIP text encoder and contrastive loss for image encoder

Previous studies primarily focus on utilizing CLIP's visual feature extraction capability while neglecting its text feature extraction capability. Therefore, as illustrated in Fig 4, *CLIP.text* is introduced to encode the textual information and help to guide the CLIP fine-tuning through $loss_{IE}$ during the training.

Given the pre-built queue $K_t = (k_t^1, \cdots, k_t^i, \cdots, k_t^N)$ which contains all [EOS] token features extracted by *CLIP.text* from the image captions in the training dataset, the i-th image's [CLS] token vector $q_v^i$ output by *CLIP.visual*, $loss_{IE}$ is defined by:

$$loss_{IE}(i) = -\log \frac{\exp(q_v^i \cdot k_t^i / \tau)}{\sum_{j=1}^{N} \exp(q_v^i \cdot k_t^j / \tau)}.$$

(2)

Here, $\tau$ is the temperature coefficient, N is the number of image-text pairs in the training dataset.

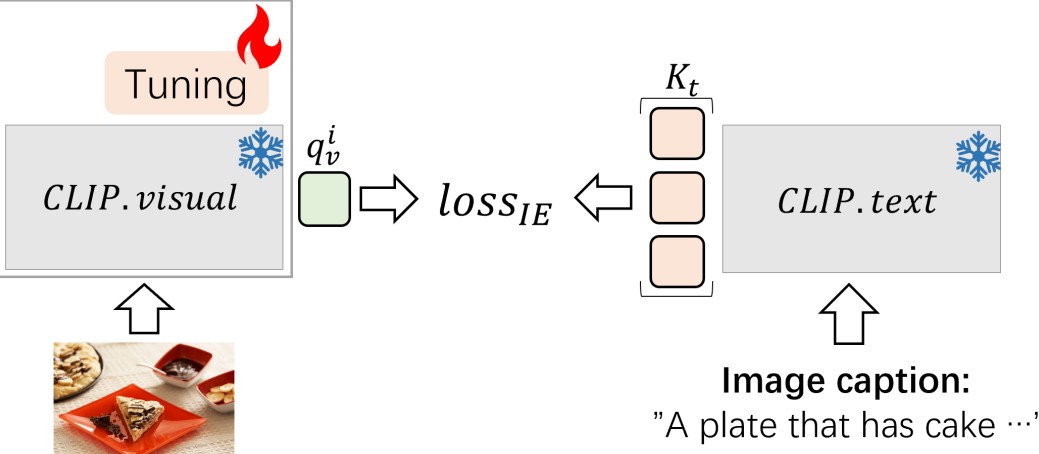

**Fig 4. *CLIP.text* and $loss_{IE}$.** The semantics spaces of *CLIP.text* and *CLIP.visual* are aligned through $loss_{IE}$. $K_t$ is the pre-built queue, which contains all [EOS] token features extracted by *CLIP.text* from the image captions in the training dataset. $q_v^i$ denotes the i-th image's [CLS] token vector output by *CLIP.visual*. In addition, $loss_{IE}$ should be used together with fine-tuning methods. If the original multi-modal LLMs do not provide the fine-tuning methods for CLIP, we can skip using it or use the prompt-tuning method by default.

## Contrastive loss for ST

Most of current multi-modal LLMs use a large number of high-quality image-text pairs for pre-training ST to align the semantic spaces between vision and language, which results in a surge in training costs. Therefore, as illustrated in Fig 5, $loss_{ST}$ is proposed to help align the multi-modal semantic spaces between vision and language with affordable training costs.

Given the pre-built queue $K_e = (k_e^1, \cdots, k_e^i, \cdots, k_e^N)$ which contains the average of the sentence token features that are extracted by $LM_e$ from the instructions and responses in the training dataset, $q_{ST}^i$ which averages the i-th image's token features output by ST, $loss_{ST}$ is defined by:

$$loss_{ST}(i) = -\log \frac{\exp(q_{ST}^i \cdot k_e^i / \tau)}{\sum_{j=1}^{N} \exp(q_{ST}^i \cdot k_e^j / \tau)}. \tag{3}$$

Here, $\tau$ is the temperature coefficient, and N is the number of image-text pairs in the training dataset.

## Contrastive loss for LLM

Unlike existing works, as illustrated in Fig 6, the contrastive loss $loss_{LM}$ is proposed to help improve the performance of LLMs, rather than solely relying on using PETL and replacing LLMs with larger ones.

Given the pre-built queue $K_b = (k_b^1, \cdots, k_b^i, \cdots, k_b^N)$ which contains all [EOS] token features extracted by the original LLM's $LM_{block}$ from the instructions and responses in the training dataset, the i-th text's last token vector $q_{LM}^i$ output by the tuning LLM's $LM_{block}$, $loss_{LM}$ is defined by:

$$loss_{LM}(i) = -\log \frac{\exp(q_{LM}^i \cdot k_b^i / \tau)}{\sum_{j=1}^{N} \exp(q_{LM}^i \cdot k_b^j / \tau)}. \tag{4}$$

Here, $\tau$ is the temperature coefficient, and N is the number of image-text pairs in the training dataset.

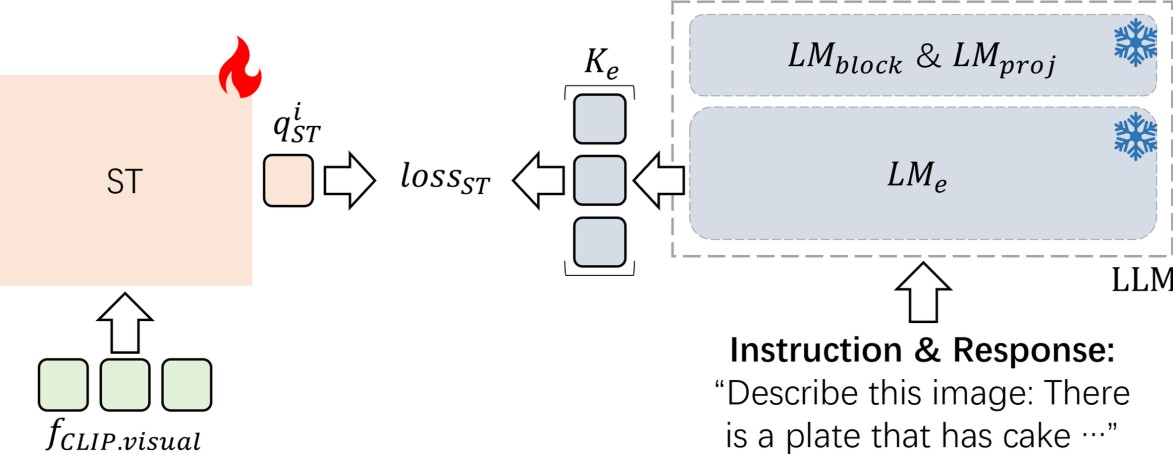

**Fig 5.** $loss_{ST}$, **which is used to align the semantic spaces of images and texts by maximizing the similarity between the** $q_{ST}^i$ **and the i-th vector of the queue** $K_e$. $K_e$ is the pre-built queue containing the average of the sentence token features, which are extracted by $LM_e$ from the instructions and responses in the training dataset. $q_{ST}^i$ is the vector that averages the token features of the i-th image output by ST. $LM_e$ represents the LLM's embedding layer. $LM_{block}$ represents the hidden layers of LLM. $LM_{proj}$ is the output layer of LLM.

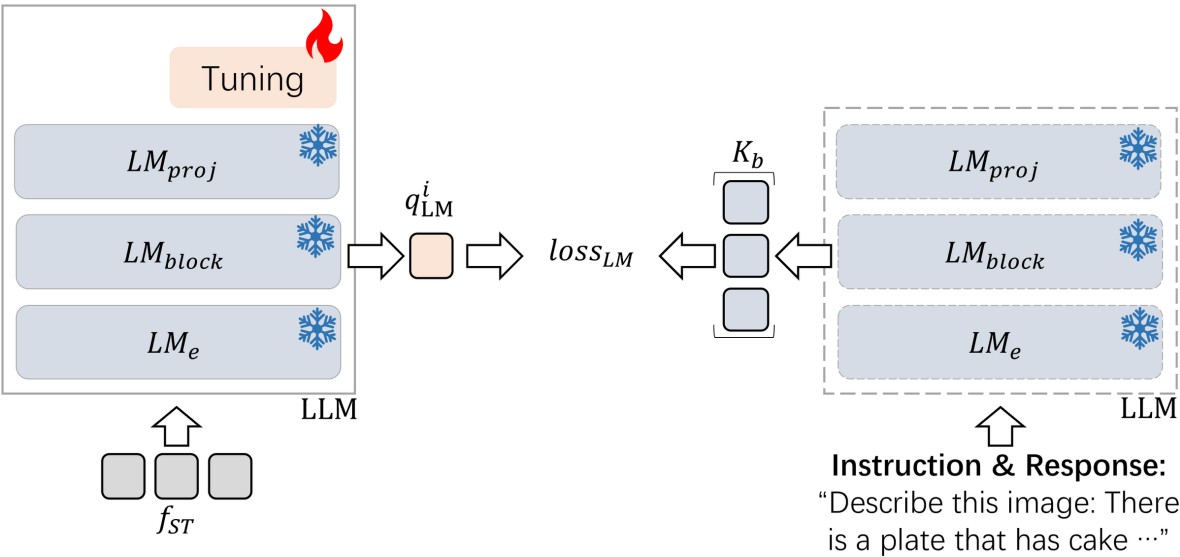

**Fig 6.** $loss_{LM}$**, which is used to align the semantic spaces of the tuning LLM and the original LLM by maximizing the similarity between the** $q^i_{LM}$ **and the i-th vector of the queue** $K_b$**.** $K_b$ is the pre-built queue, which contains all [EOS] token features extracted by the original LLM's $LM_{block}$ from the instructions and responses in the training dataset. $q^i_{LM}$ is the i-th text's last token vector output by the tuning LLM's $LM_{block}$. $LM_{block}$ represents the hidden layers of LLM. $LM_{proj}$ is the output layer of LLM.

## Segmented training

In experiments, we find that although the three contrastive losses $loss_{IE}$, $loss_{ST}$, and $loss_{LM}$ can improve the performance of multi-modal LLMs when applied individually, using them simultaneously in one training stage may lead to training instability. To avoid the training instability, we use the segmented training to isolate the influence between different losses, thus enabling the stable performance improvements when using the combination of the three contrastive losses. In each stage, the training data is the training part of the dataset. Table 1 shows the details of segmented training.

## Experiments

Following LLaMA-Adapter and LaVIN, we first apply EIM on ClipCap and evaluate the model on the traditional image captioning dataset COCO, then extend EIM to the representative multi-modal language models, such as LaVIN and LLaMA-Adapter, and evaluate the models on the first large scale multi-modal dataset ScienceQA, and finally apply EIM on LaVIN and evaluate the model on the first multi-modal LLM evaluation benchmark

**Table 1. Segmented training.**

| Stage | Queue | Objective | Tuning | | |
|---|---|---|---|---|---|
| | | | CLIP | ST | LLM |
| Stage 1 | $K_e$ | $loss_{ST}$ | × | √ | × |
| Stage 2 | $K_b$ | $loss_{LM}$ | × | × | √ |
| Stage 3 | $K_t$ | $loss_{base} + loss_{IE}$ | √ | √ | √ |

We use the segmented training to isolate the influence between different losses, thus ensuring the stability of the training process.

MME. We choose ClipCap because it still serves as the baseline in LLaMA-Adapter V2. We choose LaVIN and LLaMA-Adapter because they are representative works of the current multi-modal LLMs, with significant differences in fine-tuning methods and the ST structure, which can provide a more comprehensive test of our solution. LLaMA-Adapter does not use the fine-tuning method on CLIP, uses Transformer structure to design the ST, and uses the prompting method to fine-tune the LLM. In contrast, LaVIN uses the adapter to fine-tune CLIP, uses a lightweight MLP as ST, and uses the adapter MMA to fine-tune the LLM.

The COCO Caption dataset experimental results of ClipCap$_{eim}$, which is a model that applies EIM on the ClipCap$_{small}$, show the 1.75% performance improvement when compared to those of ClipCap$_{large}$, which has 3.13 times the number of parameters of ClipCap$_{eim}$. Furthermore, we extend EIM to the representative multi-modal LLMs, including LLaMA-Adapter and LaVIN, and evaluate on the ScienceQA dataset, achieving accuracy improvements of 2.76% and 2.05%, respectively, which confirms the effective performance improvement of EIM for multi-modal LLMs. Finally, we apply EIM on LaVIN-7B and evaluate on the MME benchmark, achieving comparable performance when compared to LaVIN-13B. The rest of this section is introduced in the order of the datasets and metrics, the implementation details, the experimental results of ClipCap on the COCO dataset, and the experimental results of two representative multi-modal LLMs on ScienceQA.

## Datasets and metrics

**COCO caption.** COCO Caption dataset [85] contains 0.6M training image caption data (120k images with 5 captions per image) over a wide range of distributions. We split the dataset according to the Karpathy [86] split. Similar to Oscar [87], we validate ClipCap over the COCO Caption dataset using the common metrics BLEU4 [88], METEOR [89], CIDEr [90], SPICE [91] and ROUGE_L [92], and the main reference metric is BLEU4.

**ScienceQA.** ScienceQA [93] is the first large-scale multi-modal dataset designed for science question answering, covering multiple fields, including 3 subjects, 26 topics, 127 categories, and 379 skills. The dataset includes pure text and text-image examples, which are divided into three parts: train, validation, and test, with 12,726, 4,241, and 4,241 examples, respectively. Following LaVIN [22] and LLaMA-Adapter [19], we evaluate the models on the ScienceQA dataset using the average accuracy.

**Alphaca-52k & LLaVA-158k & MME.** Alphaca-52k [94] contains 52k text-only instruction-following data generated by GPT-3.5 [95]. LLaVA-158k [11] is a large-scale image-text instruction dataset where the answer is automatically generated by GPT-4 [4]. Following LaVIN [22], we train the multi-modal chatbot model on Alphaca-52k & LLaVA-158k and evaluate the model on the first multi-modal LLM evaluation benchmark MME [96], which is free and widely used in 50+ recent multi-modal LLMs.

MME includes two major categories: perception and cognition. The former, with 10 subtasks, refers to recognizing specific objects in images, while the latter, with 4 subtasks, is more challenging for deducing complex answers from visual information. MME manually designs the annotations of instruction-answer pairs to avoid data leakage that may arise from the direct use of public datasets for evaluation. For each test image, MME adopts an instruction of a question and a description "Please answer yes or no", which prompts LLMs to answer "yes" or "no". The full score of MME is 2800.

## Implementation details

The performance of EIM applied to ClipCap is evaluated on the COCO Caption dataset with a single RTX-3090 graphics card. Following the settings in ClipCap, the epochs, batch size,

and learning rate are set to 10, 40 and 2E-5, respectively. In order to apply EIM to ClipCap, we make modifications to ST due to the use of the full visual features instead of a single visual feature, and use the segmented training to ensure the stability of the training process. The segmented training sequence is $loss_{ST}$, $loss_{LM}$, and $loss_{base} + loss_{IE}$. To reduce the training costs and ensure the fair comparison of experiments, only the $loss_{base} + loss_{IE}$ stage trains the entire model. In the $loss_{ST}$ stage, only the ST is trained through the training objective $loss_{ST}$. In the $loss_{LM}$ stage, only the LLM is trained through the training objective $loss_{LM}$.

The performance of EIM applied to the representative models is evaluated on the ScienceQA dataset with four A100 graphics cards. Following the settings in LaVIN and LLaMA-Adapter, the epochs, batch size, and learning rate are 20, 32, and 9E-3, respectively. Due to the adapter and prompt methods used in LaVIN and LLaMA-Adapter, which harm the training stability, a new training stage *Adapter*, which fine-tunes the multi-modal LLMs solely by $loss_{base}$, is added after the $loss_{LM}$ stage. Therefore, the segmented training sequence is $loss_{ST}$, $loss_{LM}$, *Adapter*, and $loss_{base} + loss_{IE}$.

We also apply EIM to LaVIN-7B-lite, train the chatbot model on four RTX-3090 graphics cards, and evaluate on the MME benchmark. Following LaVIN, the training settings are the same, except that the training epochs are 15 in the original LaVIN and 10 in ours.

Following the previous works [80,84], the temperature coefficient $\tau$ is set as 0.07. It is worth noting that in our solution, $loss_{IE}$ uses the image caption, $loss_{ST}$ uses the instruction and response, and $loss_{LM}$ uses the instruction and response. These losses are not applied in the inference stage, so there will be no additional restrictions on model deployment and application scenarios. In practice, the instruction is empty, and the response is caption in COCO Caption dataset. In the ScienceQA dataset, the instruction includes question, context, and options, and the response includes answer, lecture, and solution. We think that both visual question answering and image captioning can be seen as tasks for generating a response when given an image and an instruction as input, and the combination of the instruction and response can describe the image from another perspective. Specially, we observe that the solution in ScienceQA contains a lot of image related information. Therefore, if the proposed loss is effective on image captioning datasets like COCO Caption, it should also be effective on visual question answering datasets like ScienceQA.

## Experimental results on COCO

We choose the ClipCap as the baseline model and conduct experiments to quickly validate the feasibility of our ideas. ClipCap uses the CLIP VIT-B/32 as the visual encoder and uses GPT-2 as the language model. The ST in ClipCap is an MLP with a hidden layer and is used to translate a single visual token with 512 dimensions to 10 tokens with 768 dimensions. ClipCap does not provide the fine-tuning methods on CLIP and fully fine-tunes GPT-2.

From Table 2, ClipCap$_{eim}$ is the model that applies EIM on ClipCap$_{small}$, which achieves the 2.7% performance improvement in BLEU4 when compared to ClipCap$_{small}$. Furthermore, ClipCap$_{eim}$ achieves the 1.75% performance improvement with only 31.99% total parameters when compared to ClipCap$_{large}$. In practice, the parameter size of ClipCap$_{eim}$ is smaller than that of ClipCap$_{small}$ because of a certain structure change of ST due to the usage of full visual features instead of using a single visual feature in original ClipCap$_{small}$ and removing the CLIP's output layer. In the original ClipCap, the input dimension of ST is 512, the hidden dimension of ST is 3840, and the output dimension of ST is 7680. Because this ST is responsible for converting one token into multiple tokens, if we use the full visual features, the input dimension, hidden dimension and output dimension will be changed to 197376, 98688, and 197376, respectively. It is too large and has 39B parameters. So, we changed the ClipCap ST

**Table 2. The experimental results (%) of applying EIM to ClipCap on the COCO Caption dataset.**

| Methods | LMs | #Params | BLEU4 | METERO | CIDEr | SPICE | ROUGE_L |
|---------|-----|---------|-------|--------|-------|-------|---------|
| ClipCap$_{small}$ | GPT2-SMALL | 307.19M | 31.20 | 26.50 | 105.20 | 19.50 | 54.60 |
| ClipCap$_{large}$ | GPT2-LARGE | 956.78M | 32.15 | 27.10 | 108.35 | 20.12 | - |
| **ClipCap$_{eim}$** | GPT2-SMALL | 306.11M | **33.90 (+2.7)** | 27.40 | 111.90 | 20.40 | 55.90 |

ClipCap$_{eim}$ is the model that applies EIM on ClipCap$_{small}$. The bolded number indicates the best result of main metric. #Params is the number of parameters of the model.

structure to a simple FFN, with the input dimension of 768, the hidden dimension of 3072, and the output dimension of 768. The ST parameters are reduced from the original 31.47M to 4.72M. While the prompting method introduces 26.07M parameters, the number of trainable parameters is reduced by 0.68M. Finally, the total parameters is reduced by 1.08M, which is the sum of the reduced trainable parameters 0.68M and the removed CLIP's output layer parameters 0.4M. What's more, we do not need to modify the ST to the extent of ClipCap in current representative models such as LaVIN, LLaMA-Adapter. For instance, we just modify the input dimension of ST from 768 to 1024 in LLaMA-Adapter and do not need to modify the ST in LaVIN.

**Ablation study.** To demonstrate the reliability of EIM in detail, ablation experiments are conducted. As shown in Table 3, when using $loss_{ST}$ and $loss_{LM}$, the performance is increased by 0.8% and 0.6% respectively. When using $loss_{IE}$, the performance is improved by 1.6% with the increased parameters from the default prompt network. In contrast, if the original model provides a fine-tuning method on CLIP, $loss_{IE}$ can be directly used without increasing parameters. We also present the case studies in Fig 7, which show that ClipCap$_{eim}$ generates more accurate and detailed captions than ClipCap$_{small}$.

**Segmented training.** The experimental results of the segmented training are detailed in Table 4. The results show that when removing the CLIP's output layer and using the full visual features on ClipCap$_{small}$, the performance is improved by 1.7%. After introducing the $loss_{ST}$ stage, the performance is improved by 1.9%. After introducing the $loss_{LM}$ stage, the performance is improved by 2.3%. After introducing the $loss_{base} + loss_{IE}$ stage, the performance is improved by 2.7%.

**Why use segmented training?** We have tried to combine different losses, but the performance is unstable. As shown in Table 5, when we use $loss_{base}$, $loss_{ST}$, $loss_{LM}$, and $loss_{IE}$ together in one training stage, the performance is 33.0%. When we use segmented training, the performance is improved to 33.9% with a 0.9% improvement. As shown in Table 6, when we use both $loss_{ST}$ and $loss_{LM}$ simultaneously, the performace is 32. The result is lower than our

**Table 3. Ablation study (%).**

| Settings | BLEU4 | METERO | CIDEr | SPICE | ROUGE_L |
|----------|-------|--------|-------|-------|---------|
| $loss_{base}$ | 31.2 | 26.5 | 105.2 | 19.5 | 54.6 |
| + $loss_{ST}$ | **32.0 (+0.8)** | 26.7 | 107.1 | 19.8 | 54.9 |
| + $loss_{LM}$ | **31.8 (+0.6)** | 26.6 | 106.3 | 19.6 | 54.6 |
| + $loss_{IE}$ | **32.8 (+1.6)** | 26.9 | 108.6 | 20.0 | 55.2 |

$loss_{base}$ represents training the baseline model ClipCap$_{small}$ with $loss_{base}$. Each setting is applied to ClipCap$_{small}$ individually with the same training hyperparameters.

| | | | | |
|---|---|---|---|---|
| Input image: | | | | |
| $ClipCap_{small}$ | A red and white airplane flying in the air. | A blue motorcycle parked in front of a building. | A painting of a vase with a red apple and a candle. | Three ducks are swimming in a pond. |
| $ClipCap_{eim}$ | A red airplane flying in the sky with smoke coming out of it. | A row of motorcycles parked on a street. | A painting of a vase with oranges and a candle. | Two ducks are swimming in a green pond. |
| Ground truth | An airplane flying in the sky with smoke coming out of it. | A row of motorcycles parked in front of a building. | Painting of oranges, a bowl, candle, and a pitcher. | Two ducks are swimming in the green colored pond. |
| Input image: | | | | |
| $ClipCap_{small}$ | A zebra standing in the snow next to a tree. | A cow standing on a street corner next to a red car. | A cow standing in a field with a man on it. | A man riding a horse drawn cart down a dirt road. |
| $ClipCap_{eim}$ | A zebra standing in the snow next to a brick wall. | A cow walking down a street next to a store. | A cow walking through a lush green field. | A man and his dog on a dirt road. |
| Ground truth | One zebra standing in snow near a stone wall. | A cow standing near a curb in front of a store. | A cow standing in a grassy open field. | A hiker and dogs are walking in a canyon. |

**Fig 7. Case study of applying EIM to ClipCap_small on the COCO Caption dataset.** We show the caption generated by ClipCap_small, the caption generated by ClipCap_eim, and the ground truth. ClipCap_eim generates more accurate and detailed captions than ClipCap_small. Incorrect text is highlighted in red.

**Table 4. Segmented training (%).**

| Settings | BLEU4 | METERO | CIDEr | SPICE | ROUGE_L |
|---|---|---|---|---|---|
| $loss_{base}$ | 31.2 | 26.5 | 105.2 | 19.5 | 54.6 |
| $loss_{base} + model_{opt}$ | **32.9 (+1.7)** | 27.1 | 110 | 20.1 | 55.4 |
| $loss_{ST} \rightarrow loss_{base}$ | **33.1 (+1.9)** | 27.3 | 110.4 | 20.4 | 55.6 |
| $loss_{ST} \rightarrow loss_{LM} \rightarrow loss_{base}$ | **33.5 (+2.3)** | 27.4 | 111.1 | 20.4 | 55.8 |
| $loss_{ST} \rightarrow loss_{LM} \rightarrow loss_{base} + loss_{IE}$ | **33.9 (+2.7)** | 27.4 | 111.9 | 20.4 | 55.9 |

The first row is the performance of the baseline model ClipCap_small. $model_{opt}$ represents modifying the usage of CLIP features, including removing the output layer and using full visual features in ClipCap. Each setting is applied sequentially.

**Table 5. Experimental results (%) of applying $loss_{ST}$, $loss_{LM}$, and $loss_{IE}$ in one training stage and segmented training.**

| Settings | BLEU4 |
|---|---|
| $loss_{base} + loss_{ST} + loss_{LM} + loss_{IE}$ | 33.0 |
| $loss_{ST} \rightarrow loss_{LM} \rightarrow loss_{base} + loss_{IE}$ | **33.9 (+0.9)** |

expectation. We speculate that this is due to the inconsistent optimization goals among the different losses. $loss_{IE}$ is designed to help the visual encoder to adapt to downstream tasks, $loss_{ST}$ is designed to help ST align the image and text, and $loss_{LM}$ is designed to preserve the performance of LLM. Therefore, we completely eliminate the interference between these three

**Table 6. Experimental results (%) with the combination of $loss_{ST}$ and $loss_{LM}$.**

| Settings | BLEU4 |
|---|---|
| $loss_{base} + loss_{ST}$ | 32.0 |
| $loss_{base} + loss_{LM}$ | 31.8 |
| $loss_{base} + loss_{ST} + loss_{LM}$ | 32.0 |

losses through gradient truncation and segmented training to achieve stable performance improvement.

We refer to the common training approach of multi-modal LLMs, that is firstly pre-training the ST and then fine-tuning the LLM. $loss_{IE}$ is placed in the last as a supplement because models like LLaMA-Adpater do not fine-tune the CLIP. We also conduct experiments to ensure the order of these losses. As shown in Table 7, we conduct experiments to verify whether the pre-training order of $loss_{ST}$ and $loss_{LM}$ is reasonable. We also verify whether $loss_{IE}$, as a supplement, should be placed at the beginning or at the end of the training sequence, and the results are shown in Table 8. In practice, we think $loss_{IE}$ should be used together with $loss_{base}$ to prevent the excessively fine-tuning of CLIP. We can only provide such a stable version right now. We will try to integrate these losses in future work.

**Modifying the usage of CLIP features.** In this paper, modifying the usage of CLIP features includes removing CLIP's output layer and using the full visual features. According to Table 9, when removing the CLIP's output layer or using the full visual features alone, the performance is 31.6% and 31.8%, respectively. However, when both are used simultaneously, the performance is improved to 32.9%. We speculate that the visual features after removing the output layer contain more information, which benefits the use of the full visual features. Our findings indirectly confirm the speculation that the use of the output layer of CLIP is detrimental to downstream task training due to the differences between the pre-training task and downstream tasks.

**Table 7. Experimental results (%) under different orders of $loss_{ST}$ and $loss_{LM}$.**

| Settings | BLEU4 |
|---|---|
| $loss_{ST} \rightarrow loss_{LM} \rightarrow loss_{base}$ | **33.5** |
| $loss_{LM} \rightarrow loss_{ST} \rightarrow loss_{base}$ | 33.1 |

**Table 8. Experimental results (%) of $loss_{IE}$ at both ends of the sequence.**

| Settings | BLEU4 |
|---|---|
| $loss_{IE} \rightarrow loss_{ST} \rightarrow loss_{LM} \rightarrow loss_{base}$ | 33.4 |
| $loss_{ST} \rightarrow loss_{LM} \rightarrow loss_{base} + loss_{IE}$ | **33.9** |

**Table 9. Experimental results (%) of modifying the usage of CLIP features.**

| Settings | BLEU4 |
|---|---|
| $loss_{base}$ + removing the CLIP's output layer | 31.6 |
| $loss_{base}$ + using full visual features | 31.8 |
| $loss_{base}$ + both | 32.9 |

The CLIP's output layer is a linear layer used to change the hidden dimension to the output dimension. The full visual features are the [CLS] token and the patch tokens extracted by CLIP.

**Comparison of training efficiency.** According to Table 2, EIM can improve the performance with fewer parameters, which means that there will be no additional hardware overhead. However, due to the segmented training method used in this paper, the training time will be increased compared to ClipCap$_{small}$. One epoch training time for ClipCap$_{small}$, ClipCap$_{large}$, and ClipCap$_{eim}$ is 23 minutes, 75 minutes, and 51 minutes, respectively. The training time for the three stages of segmented training is 0.9 minutes, 6.4 minutes, and 43.7 minutes, respectively. The increase in training time mainly comes from the $loss_{base}+loss_{IE}$ stage due to differences in CLIP usage. In ClipCap$_{eim}$, $loss_{IE}$ necessitates the use of CLIP during the training phase. In contrast, ClipCap uses pre-extracted visual features from CLIP instead of using CLIP during the training phase. This is confirmed in subsequent LLaMA-Adapter and LaVIN model experiments. LLaMA-Adapter and LaVIN use CLIP during the training phase instead of pre-extracted visual features, which does not increase training time during the $loss_{base} + loss_{IE}$ training phase.

## Results on ScienceQA

We choose LaVIN and LLaMA-Adapter because they are representative works of the current multi-modal LLMs, with significant differences in fine-tuning methods and the ST structure, which can provide a more comprehensive test of our solution. LLaMA-Adapter uses the CLIP VIT-L/14 as the visual encoder, uses the Transformer structure to design the ST, and uses the LLaMA as the LLM. In terms of fine-tuning methods, LLaMA-Adapter does not fine-tune CLIP and fine-tunes the LLM through the prompting method. In contrast, LaVIN uses the CLIP VIT-L/14 as the visual encoder, uses a lightweight MLP as ST, and uses the LLaMA as the LLM. In terms of fine-tuning methods, LaVIN uses the adapter to fine-tune CLIP and LLM.

As shown in Tables 10 and 11, we extend EIM to the representative multi-modal LLMs, including LLaMA-Adapter and LaVIN, and evaluate on the ScienceQA dataset, achieving accuracy improvements of 2.76% and 2.05%, respectively, which confirms the effective performance improvement of EIM for multi-modal LLMs. It should be noted that because LaVIN takes the output features of ST as the input of the LLM and there is a maximum input length limitation in LaVIN's LLM setting, we choose 50 visual token vectors instead of using the full visual features. In contrast, LLaMA-Adapter uses the full visual features.

**Results on LLaMA-adapter.** Table 10 shows the experimental results of applying EIM on LLaMA-Adapter. From the table, it can be seen that the performance improved by 2.76%. However, due to the default prompt tuning method, the number of trainable parameters is increased by nearly 55M. Therefore, we also provide the experimental results for only the first

**Table 10. The experimental results (%) of applying EIM to LLaMA-Adapter on the ScienceQA dataset.**

| Methods | LLMs | Subject | | | Context Modality | | | Grade | | Average |
|---|---|---|---|---|---|---|---|---|---|---|
| | | NAT | SOC | LAN | TXT | IMG | NO | G1-6 | G7-12 | |
| LLaMA-Adapter | LLaMA-7B | 84.37 | 88.30 | 84.36 | 83.72 | 80.32 | 86.90 | 85.83 | 84.05 | 85.19 |
| **Ours**($loss_{ST} \rightarrow loss_{LM}$ $\rightarrow Adapter_{LM}$) | LLaMA-7B | 86.63 | 94.26 | 84.27 | 86.12 | 85.42 | 86.62 | 88.07 | 86.82 | **87.62 (+2.43)** |
| **Ours**($loss_{ST} \rightarrow loss_{LM}$ $\rightarrow Adapter_{LM} \rightarrow$ $loss_{base} + loss_{IE}$) | LLaMA-7B | 87.08 | 93.36 | 85.36 | 86.41 | 84.93 | 87.74 | 88.07 | 87.74 | **87.95 (+2.76)** |

LLaMA-Adapter is the baseline model. Question classes: NAT = natural science, SOC = social science, LAN = language science, TXT = text context, IMG = image context, NO = no context, G1-6 = grades 1-6, G7-12 = grades 7-12.

**Table 11. The experimental results (%) of applying EIM to LaVIN on the ScienceQA dataset.**

| Methods | LLMs | Subject | | | Context Modality | | | Grade | | Average |
|---|---|---|---|---|---|---|---|---|---|---|
| | | NAT | SOC | LAN | TXT | IMG | NO | G1-6 | G7-12 | |
| LaVIN-7B-lite(llama) | LLaMA-7B | 88.37 | 94.38 | 84.18 | 87.49 | 86.42 | 87.60 | 89.39 | 87.01 | 88.35 |
| LaVIN-7B (vicuna) | Vicuna-7B | 89.25 | 94.94 | 85.24 | 88.51 | 87.46 | 88.08 | 90.16 | 88.07 | 89.41 |
| LaVIN-7B (llama) | LLaMA-7B | 89.17 | 94.94 | 85.64 | 88.37 | 87.46 | 88.36 | 90.64 | 87.34 | 89.46 |
| LaVIN-13B-lite(llama) | LLaMA-13B | 89.34 | 91.00 | 87.55 | 88.12 | 85.28 | 90.10 | 89.76 | 88.27 | 89.22 |
| LaVIN-13B(llama) | LLaMA-13B | 90.01 | 94.60 | 88.36 | 89.05 | 87.31 | 90.80 | 91.34 | 89.12 | 90.54 |
| **LaVIN-7B-lite\*$_{eim}$** | LLaMA-7B | 89.17 | **94.94** | 87.73 | 88.37 | 87.21 | 90.03 | **91.49** | 89.12 | **90.00 (+1.65)** |
| **LaVIN-7B-lite$_{eim}$** | LLaMA-7B | 89.74 | 94.60 | **88.36** | **88.91** | **87.51** | **90.87** | 90.75 | **89.78** | **90.40 (+2.05)** |

Our model is based on LaVIN-7B-lite(llama). LaVIN-7B-lite\*$_{eim}$ is the model that applies EIM on LaVIN-7B-lite(llama) with the LLM adapter network trained in the *Adapter* stage. LaVIN-7B-lite$_{eim}$ is the model that applies EIM on LaVIN-7B-lite(llama) with the LLM adapter network and the CLIP adapter network trained in the *Adapter* stage. LaVIN-7B-lite(llama) represents that the number of the parameters is 7B, the LLM is LLaMA, and the suffix word lite means using 4-bit training. The bolded numbers indicate the best results.

three stages of training, which can improve performance by 2.43% without increasing the number of parameters. It is worth noting that when we apply $loss_{ST}$ and $loss_{LM}$, the performance of all tasks has significantly increased except for LAN and NO. After $loss_{IE}$ is applied, the performance of LAN and NO tasks is increased by 1.01% and 1.12%, respectively, which indicates that $loss_{IE}$ and the other two losses are complementary. Besides, the G7-12 task is increased from 86.82% to 87.74%, which indicates that $loss_{IE}$ can further improve the complex reasoning ability. However, we also notice that there is a little performance decrease on IMG and SOC tasks which rely more on the image understanding ability. We think this is because the ScienceQA dataset contains text-only samples which are better not to be used in $loss_{IE}$.

**Results on LaVIN.** From Table 11, LaVIN-7B-lite$_{eim}$ and LaVIN-7B-lite\*$_{eim}$ can achieve optimal or suboptimal performance on almost all tasks even when compared to the best model LaVIN-13B. LaVIN-7B-lite$_{eim}$ achieves the 2.05% performance improvement. It does not increase the number of model parameters and trainable parameters when compared to LaVIN-7B-lite(llama) because that LaVIN provides the Adapter to fine-tune the CLIP. Furthermore, LaVIN-7B-lite$_{eim}$ achieves the 1.18% performance improvement with only 7B total parameters when compared to the 13B parameters model LaVIN-13B-lite(llama). In addition, we also provide the experimental results of LaVIN-7B-lite\*$_{eim}$ that applies EIM on LaVIN-7B-lite(llama) with the LLM adapter network trained in the *Adapter* stage, which can improve the performance by 1.65%. Compared to the 13B model LaVIN-13B-lite(llama), LaVIN-7B-lite\*$_{eim}$ can also achieve the 0.78% performance improvement. The improvement of EIM on LaVIN is weaker than that applied to ClipCap and LLaMA-Adapter. We speculate this is mainly due to the fact that we only use the 20% visual features instead of using the full visual features.

**Comparison of training efficiency.** The training time of one epoch is increased by 8 minutes after applying EIM to LLaMA-Adapter, with 20 seconds from the $loss_{ST}$ stage, 1 minute from the $loss_{LM}$ stage and nearly 7 minutes from the *Adapter* stage. After applying EIM to LaVIN, the training time of one epoch is increased by 8.5 minutes, with 27 seconds, 1.05 minutes, and 7 minutes from the $loss_{ST}$ stage, the $loss_{LM}$ stage, and the *Adapter* stage, respectively. Introducing $loss_{IE}$ in the $loss_{base} + loss_{IE}$ stage does not increase the training time in both experiments, which confirm the analysis that the increase time in the ClipCap $loss_{base} + loss_{IE}$ stage is due to the different use of CLIP. Although the training time on both experiments is increased by nearly 120%, primarily due to the *Adapter* stage, we believe that this is acceptable when compared to the performance improvements.

## Experimental results on MME

In this paper, we apply EIM on LaVIN-7B-lite, follow the original LaVIN model [22] to train the chatbot model on Alphaca-52k & LLaVA-158k datasets, and evaluate on the first multi-modal LLM evaluation benchmark MME [96].

As shown in Tables 12 and 13, our model LaVIN-7B-lite$_{eim}$ can achieve comparable performance in Perception and Cognition tasks when compared to LaVIN-13B. It can be seen that LaVIN-7B-lite$_{eim}$ can achieve significantly better performance in Celebrity, Scene, Landmark, Artwork, and Text Translation tasks, and comparable performance in Position, Color, Poster, Commonsense Reasoning, and Code Reasoning tasks. This shows that our solution EIM can improve the model performance from the image understanding and complex reasoning abilities and preserve the knowledge of LLM. For instance, when compared to LaVIN-13B, LaVIN-7B-lite$_{eim}$ can archive 32.36, 7.25, and 4.5 improvements in Celebrity, Landmark, and Artwork tasks, which rely more on image understanding ability and the knowledge of LLM. And LaVIN-7B-lite$_{eim}$ can achieve 14.5 improvement in the Scene task which relies more on the image understanding ability. Besides, LaVIN-7B-lite$_{eim}$ can achieve 10 improvements in the Text Translation task which relies more on complex reasoning ability. However, the performance of LaVIN-7B-lite$_{eim}$ on Existence, Count, and OCR tasks is lower than that of LaVIN-13B. We speculate this is due to the fact that we only use 20% visual features in LaVIN-7B-lite$_{eim}$ rather than using full visual features. This will reduce the model performance in Existence, Count, and OCR tasks which partial visual features have a significant impact on results. So, we think we can introduce more partial visual features in future works to improve the performance of LaVIN-7B-lite$_{eim}$ on tasks like OCR, Count and Existence that are sensitive to local image details. This will also benefit tasks like Position which require the local image details to better understand the image.

**Table 12. Perception performance comparison on MME benchmark.**

| Methods | Existence | Count | Position | Color | Poster | Celebrity | Scene | Landmark | Artwork | OCR | Total |
|---|---|---|---|---|---|---|---|---|---|---|---|
| LLaVA [11] | 50.00 | 50.00 | 50.00 | 50.00 | 50.00 | 48.82 | 50.00 | 50.00 | 49.00 | 50.00 | 502.82 |
| MiniGPT-4 [13] | 68.33 | 55.00 | 43.33 | 75.00 | 41.84 | 54.41 | 71.75 | 54.00 | 60.50 | 57.50 | 581.66 |
| PandaGPT [16] | 70.00 | 50.00 | 50.00 | 50.00 | 76.53 | 57.06 | 118.00 | 69.75 | 51.25 | 50.00 | 642.59 |
| Multimodal-GPT [18] | 61.67 | 55.00 | 58.33 | 68.33 | 57.82 | 73.82 | 68.00 | 69.75 | 59.50 | 82.50 | 654.72 |
| ImageBind-LLM [28] | 128.33 | 60.00 | 46.67 | 73.33 | 64.97 | 76.47 | 113.25 | 62.00 | 70.75 | 80.00 | 775.77 |
| LaVIN-13B [22] | 185.00 | 88.33 | 63.33 | 75.00 | 79.59 | 47.35 | 136.75 | 93.50 | 87.25 | 107.50 | 963.60 |
| LaVIN-7B-lite$_{eim}$ | 155.00 | 51.67 | **66.67** | **75.00** | **78.57** | **79.71** | **151.25** | **100.75** | **91.75** | 65.00 | 915.36 |

The full score of each subtask is 200, and the total score is 2000. The bolded numbers indicate the best results.

**Table 13. Cognition performance comparison on MME benchmark.**

| Methods | Commonsense Reasoning | Numerical Calculation | Text Translation | Code Reasoning | Total |
|---|---|---|---|---|---|
| LLaVA | 57.14 | 50.00 | 57.40 | 50.00 | 214.64 |
| MiniGPT-4 | 59.29 | 45.00 | 0.00 | 40.00 | 144.29 |
| PandaGPT | 73.57 | 50.00 | 57.50 | 47.50 | 228.57 |
| Multimodal-GPT | 49.29 | 62.50 | 60.00 | 55.00 | 226.79 |
| ImageBind-LLM | 48.57 | 55.00 | 50.00 | 60.00 | 213.57 |
| LaVIN-13B | 87.14 | 65.00 | 47.50 | 50.00 | 249.64 |
| LaVIN-7B-lite$_{eim}$ | 77.86 | 47.50 | **57.50** | 50.00 | 232.86 |

The full score of each subtask is 200, and the total score is 800. The bolded numbers indicate the best results.

## Conclusion

In this paper, we propose a novel effective solution, EIM, for improving the performance of multi-modal large language models from the perspective of the training process. EIM contains three losses: $loss_{IE}$, $loss_{ST}$, and $loss_{LM}$. The three losses are proposed for CLIP, ST, and LLM, respectively. $loss_{IE}$ is designed to help the visual encoder to adapt to downstream tasks, $loss_{ST}$ is designed to help ST align the image and text, and $loss_{LM}$ is designed to preserve the performance of LLM. These losses can be used separately, and we further provide a stable solution with segmented training to combine these losses. Especially if the original model, like LLaMA-Adapter, does not fine-tune the CLIP, we can omit $loss_{IE}$ to achieve 2.43% improvement or use $loss_{IE}$ to achieve further 2.76% improvement with certain new trainable parameters introduced as needed.

To validate EIM, we first apply it to ClipCap$_{small}$ and conduct experiments on the COCO Caption dataset. The experimental results show that we can achieve the 1.75% performance improvement with only 31.99% total parameters when compared to those of ClipCap$_{large}$. Secondly, we extend EIM to the multi-modal LLMs, such as LLaMA-Adapter and LaVIN, and evaluate them on the ScienceQA dataset. Finally, we also conduct multi-modal chatbot experiments with the EIM-enhanced LaVIN and evaluate it on the MME benchmark. The experimental results on the ScienceQA dataset and MME benchmark show that EIM can achieve competitive performance with 7B model parameters when compared to the 13B multi-modal LLMs, which confirms the effective performance improvement of EIM for multi-modal LLMs. However, the segmented training, which is used to ensure the stability of the training process, inevitably leads to the increase of the training time and may limit the performance. We will make improvements in future work.

## Acknowledgments

We thank the suggestions of the all anonymous reviewers.

## Author contributions

**Conceptualization:** Yuting Bai, Zixing Bai.

**Data curation:** Yuting Bai.

**Formal analysis:** Yuting Bai.

**Investigation:** Yuting Bai, Zixing Bai.

**Methodology:** Yuting Bai.

**Project administration:** Yuting Bai, Zixing Bai.

**Resources:** Yuting Bai.

**Software:** Yuting Bai, Zixing Bai.

**Supervision:** Yuting Bai.

**Validation:** Yuting Bai.

**Visualization:** Yuting Bai.

**Writing – original draft:** Yuting Bai, Zixing Bai.

**Writing – review & editing:** Yuting Bai, Tonghua Su, Zixing Bai.

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
