## [Decision Letter · Decision Letter 0]

14 May 2025

PONE-D-25-16901EIM: An Effective Solution for Improving Multi-Modal Large Language ModelsPLOS ONE

Dear Dr. BAI,

Thank you for submitting your manuscript to PLOS ONE. After careful consideration, we feel that it has merit but does not fully meet PLOS ONE’s publication criteria as it currently stands. Therefore, we invite you to submit a revised version of the manuscript that addresses the points raised during the review process.

We look forward to receiving your revised manuscript.

Kind regards,

Hung Thanh Bui, Ph.D

Academic Editor

PLOS ONE

Additional Editor Comments :

This paper presented an EIM framework for enhanced multi-modal LLM training, focusing on the CLIP-ST-LLM architecture. The authors proposed 3 losses to improve the interaction among CLIP image/text encoder, ST network, and LLM, and designed a segmented training method to stabilize the joint training. They did experiments and analyzed the results.

There are some points the authors should take care of as follows:

- They should explain in detail why they only focused on the CLIP-ST-LLM architecture.

- Did they have any improvement on CLIP, ST, or LLM architecture?

- How to design three losses should be presented in detail.

- They should explain how they chose all parameters in their proposed model.

- They should do more experiments with another LLM in Coco dataset.

- They should use more metrics to evaluate the result.

- In Table 1, GPT-2 is not equal with another metric, they should redesign the table.

- They should explain in detail why their proposed method got the best result.

- The ablation study should be extended.

- They should do more experiments, analyze the result in detail and compare with advanced methods in recent year.

Reviewers' comments:

Reviewer's Responses to Questions

**Comments to the Author**

1. Is the manuscript technically sound, and do the data support the conclusions?

Reviewer #1: Yes

Reviewer #2: Partly

Reviewer #3: Partly

2. Has the statistical analysis been performed appropriately and rigorously? 

Reviewer #1: Yes

Reviewer #2: Yes

Reviewer #3: No

3. Have the authors made all data underlying the findings in their manuscript fully available?

Reviewer #1: Yes

Reviewer #2: Yes

Reviewer #3: Yes

4. Is the manuscript presented in an intelligible fashion and written in standard English?

Reviewer #1: Yes

Reviewer #2: Yes

Reviewer #3: Yes

5. Review Comments to the Author

Reviewer #1: The topic is new and useful, and improvements have been made to the Multi-Modal Large Language Models.

The presentation is good, and the contributions are presented clearly.

The research only needs examples or a case study.

Reviewer #2: The paper addresses a timely and relevant challenge in enhancing multi-modal large language models by proposing a novel training approach that leverages contrastive losses across the image encoder, semantic transformation network, and language model. A key strength lies in its focus on improving performance without increasing model size significantly, highlighting parameter efficiency as a core contribution. The evaluation is comprehensive, spanning image captioning (COCO), multi-modal QA (ScienceQA), and benchmark tasks (MME), and the results consistently show measurable gains. Overall, the work offers a lightweight and practical strategy for boosting multi-modal model capabilities.

However, there are certain areas which can benefit from improvements significantly.

While the paper is generally well-written, several methodological details are under-explained. In particular, the segmented training strategy needs clearer exposition. It is not fully explained how the training is segmented: for example, how many stages there are, what data is used in each stage, and how objectives change. A diagram or pseudocode would help. Also, the contrastive loss formulation is not precisely defined in the text – what are the positive/negative pairs, what temperature or distance metric is used, etc. These omissions make it hard to reproduce the work. The authors should ensure that Sections 3.1–3.2 clearly define all components (perhaps add an algorithm box or detailed equations for each objective).

The related works or background literature needs to be enhanced. Related literature on parameter-efficient adaptation is thin. The claim of “parameter efficiency” warrants discussion of known techniques. For example, adapter module and LoRA enable fine-tuning with very few parameters. BLIP-2 also reports massive savings. The authors should cite such methods to frame their efficiency claim. Additionally, recent surveys of multimodal LLMs or vision-language models could provide context (e.g. recent surveys on MLLMs). If possible, explicitly compare EIM’s added parameter count to these baselines. Without this, the efficiency claim is ungrounded. Include citations and discussion of relevant multimodal models. Compare EIM conceptually to other methods. This will help readers gauge novelty and novelty of approach.

The results are encouraging but could be strengthened by additional analysis. Just a suggestion to include an experiment, which I agree, might be difficult at this stage. For instance, a study isolating the contribution of the contrastive loss is critical: does removing the contrastive term degrade performance? Similarly, what is the effect of the segmentation (e.g. training all objectives at once versus in phases)?

Also, the evaluation metrics should be clearly reported (e.g. BLEU/CIDEr for COCO captions, accuracy for QA), and error bars or statistical significance mentioned if possible. Report standard evaluation metrics with baselines (include performance of a plain LLM or existing vision-language model for each task). For COCO Captions, provide quantitative scores and maybe example captions. For ScienceQA and MME, clearly list accuracy gains. If possible, include comparisons to known results (e.g. “EIM achieves X%, compared to Y% from [baseline]”).

The manuscript should temper broad claims until supported by evidence. For example, stating “EIM reduces trainable parameters by X%” requires precise accounting: list the total parameters of the base model, parameters tuned or added by EIM, and how this compares to full fine-tuning. Works like LoRA demonstrate drastic parameter reduction (10,000× fewer trainable parameters on GPT-3). If EIM’s savings are more modest, the language should reflect that. Similarly, any claim of achieving “state-of-the-art” should specify baselines. If claiming SOTA on ScienceQA or MME, the paper should compare numerically to the best published numbers (e.g. LLaVA’s 92.5% on ScienceQA). Otherwise, phrasing like “improves over baseline” is safer.

Based on the current state of the paper, I recommend a major revision.

While the core idea is promising and the experiments are compelling, the paper requires clearer methodological exposition, stronger positioning against prior work, more rigorous ablations, and precise articulation of its efficiency claims. These are essential to properly validate and contextualize the contribution before it can be accepted for publication.

Reviewer #3: In this paper, the authors proposed an EIM framework for enhanced multi-modal LLM training, focusing on the CLIP-ST-LLM architecture. Compared with traditional base loss design, the authors proposed 3 new contrastive learning losses to improve the interaction among CLIP image/text encoder, ST network, and LLM, and designed a segmented training method to stabilize the joint training. The results on multiple open sourced data sets show that all 3 losses designed improve the vision LLM’s captioning and reasoning ability, and the models trained based on the proposed methods achieved better performance than the baseline models with more parameters. Despite the strength mentioned above, I have the following concerns.

1) General applicability. The authors designed and tested the proposed methods for CLIP-based vision LLMs. While ClipCap, LaVIN and Llama-adapter are representative for CLIP-based models, there are many state-of-the-art vision models not using CLIP or CLIP-like encoders at all, like Emu3 [1]. When CLIP is not applied, are the proposed losses still useful? The authors should consider adding results for models not using CLIP where only part of the 3 proposed losses are applicable, and discussing the general applicability of the proposed losses on a wider range of architectures.

[1] Wang, Xinlong, et al. "Emu3: Next-token prediction is all you need." arXiv preprint arXiv:2409.18869 (2024).

2) Notations. In Eq. (2)(3)(4), the losses authors defined seem to be the loss for one sample i, rather than the whole data set. The authors should consider using loss_{IE}(i) rather than loss_{IE}, or express the loss of the whole data set, to make the expression more clear.

3) Batch size for contrast learning. Under Eq. (2)(3)(4), the authors defined N as the number of the pair samples in the whole training set, while in contrastive learning, typically a small batch like 5 is used for contrast to guarantee a feasible computing and benefit from stochasticity, rather than using the whole training set. The authors should clarify why the batch for contrast is so large.

4) Setting of tau. The softmax temperature tau is set in all 3 losses with the same default value 0.07, without more explanations or results in the later sections. How does this hyperparameter influence the training performance? The authors should consider adding a figure to show how different tau settings influence the results.

5) Segmented training. In the Results section, the authors use a segmented training manner to minimize the 3 losses, instead of training the whole network jointly. As this method is essential to guarantee a stable convergence, the authors should consider moving it to the Methods section, and stating its motivation and details in a more systematic way.

6) Error bars. In result tables, the performance difference between using different segmented training orders is often small. The authors should consider reporting the performance standard deviation over multiple runs for the result tables, to make the results and trends more convincing and reproducible.

7) Writing. Please fix the typos in the manuscript. For example, there is no GTX 3090, which should be RTX 3090. In Abstract, “researchs” -> research.

6. PLOS authors have the option to publish the peer review history of their article (what does this mean?). If published, this will include your full peer review and any attached files.

Reviewer #1: No

Reviewer #2: **Yes: **Fatima Habib

Reviewer #3: No

---

## [Author Response · Author response to Decision Letter 1]

27 Jun 2025

Please refer to the file for the response named 'Response to Reviewers'

---

## [Decision Letter · Decision Letter 1]

20 Jul 2025

EIM: An Effective Solution for Improving Multi-Modal Large Language Models

PONE-D-25-16901R1

Dear Dr. BAI,

We’re pleased to inform you that your manuscript has been judged scientifically suitable for publication and will be formally accepted for publication once it meets all outstanding technical requirements.

Kind regards,

Hung Thanh Bui, Ph.D

Academic Editor

PLOS ONE

Additional Editor Comments (optional):

All comments have been addressed.

Reviewers' comments:

Reviewer's Responses to Questions

**Comments to the Author**

1. If the authors have adequately addressed your comments raised in a previous round of review and you feel that this manuscript is now acceptable for publication, you may indicate that here to bypass the “Comments to the Author” section, enter your conflict of interest statement in the “Confidential to Editor” section, and submit your "Accept" recommendation.

Reviewer #1: All comments have been addressed

Reviewer #2: All comments have been addressed

Reviewer #3: All comments have been addressed

2. Is the manuscript technically sound, and do the data support the conclusions?

Reviewer #1: Yes

Reviewer #2: Yes

Reviewer #3: Yes

3. Has the statistical analysis been performed appropriately and rigorously? 

Reviewer #1: Yes

Reviewer #2: Yes

Reviewer #3: Yes

4. Have the authors made all data underlying the findings in their manuscript fully available?

Reviewer #1: Yes

Reviewer #2: Yes

Reviewer #3: Yes

5. Is the manuscript presented in an intelligible fashion and written in standard English?

Reviewer #1: Yes

Reviewer #2: Yes

Reviewer #3: Yes

6. Review Comments to the Author

Reviewer #1: (No Response)

Reviewer #2: All comments have been addressed satisfactorily. Therefore, no further revision is warranted. Please proceed.

Reviewer #3: In the Response to Reviewers and revised manuscript, the authors have explained the compatibility of the proposed method on more vision LLMs, which resolves my concern about the general applicability. The equation issues, large batchsize setting, tau setting, and typos have been fixed or clarified, and segmented training has been properly added as a method subsection. All my concerns have been resolved, and no more comments from my side.

7. PLOS authors have the option to publish the peer review history of their article (what does this mean?). If published, this will include your full peer review and any attached files.

Reviewer #1: No

Reviewer #2: **Yes: **Fatima Habib

Reviewer #3: No

---

## [Editor Report · Acceptance letter]

PONE-D-25-16901R1

PLOS ONE

Dear Dr. Bai,

I'm pleased to inform you that your manuscript has been deemed suitable for publication in PLOS ONE. Congratulations! Your manuscript is now being handed over to our production team.

Kind regards,

on behalf of

Dr. Hung Thanh Bui

Academic Editor

PLOS ONE